# Qualitative interview study of antibiotics and self-management strategies for respiratory infections in primary care

Lisa McDermott,[1] Geraldine M Leydon,[2] Amy Halls,[2] Jo Kelly,[2] Amanda Nagle,[2] Jennifer White,[2] Paul Little,[2] The PIPS investigators

LMD and GML contributed equally.

[1]Department of Primary Care and Public Health Sciences, King's College London, London, UK
[2]Department of Primary Care and Population Sciences, Faculty of Medicine, University of Southampton, Southampton, UK

**Correspondence to**
Professor Geraldine M Leydon;
gerry@soton.ac.uk

## ABSTRACT

**Objective** To explore perceptions of illness, the decisions to consult and the acceptability of delayed antibiotic prescriptions and self-help treatments for respiratory tract infections (RTIs).

**Design** Qualitative semistructured interview study.

**Setting** UK primary care.

**Participants** 20 adult patients who had been participating in the 'PIPS' (Pragmatic Ibuprofen Paracetamol and Steam) trial in the South of England.

**Method** Semistructured telephone interviews were conducted with participants to explore their experiences and views on various treatments for RTI.

**Results** Participants had concerns about symptoms that were not clinically serious and were mostly unaware of the natural history of RTIs, but were aware of the limitations of antibiotics and did not expect them with every consultation. Most viewed delayed prescriptions positively and had no strong preference over which technique is used to deliver the delayed antibiotic, but some patients received mixed messages, such as being told their infection was viral then being given an antibiotic, or were sceptical about the rationale. Participants disliked self-help treatments that involved taking medication and were particularly concerned about painkillers in combination. Steam inhalation was viewed as only moderately helpful for mild symptoms.

**Conclusion** Delayed prescribing is acceptable no matter how the delay is operationalised, but explanation of the rationale is needed and care taken to minimise mixed messages about the severity of illnesses and causation by viruses or bacteria. Better access is needed to good natural history information, and the signs and symptoms requiring or not requiring general practitioner advice. Significant concerns about paracetamol, ibuprofen and steam inhalation are likely to need careful exploration in the consultation.

## BACKGROUND

The overuse of antibiotics can contribute to the spread of resistant bacteria.[1 2] This problem is currently on the increase and has been identified by WHO as a serious issue that must be addressed with urgency.[3]

### Strengths and limitations of this study

► A range of views were solicited and data saturation was reached.
► Where there is existing literature, this study resonates with it.
► The interviews permitted open exploration of delayed prescribing, which is novel.
► Interviews are reports and not a window onto the actual events/experiences reported.
► Trial participation may have led to sample of participants particularly interested in this research and may not represent views of 'typical' patients.

Despite evidence to suggest the limited benefit of antibiotics in the treatment of respiratory tract infections (RTIs) in particular,[4 5] RTIs account for >60% of all antibiotic prescriptions in primary care[6] and primary care accounts for 80% of human antibiotic consumption.[7] The delayed prescribing of antibiotics is a technique that may help to reduce unnecessary prescribing in primary care.[8 9] The method is recommended by the National Institute for Health and Care Excellence (NICE) guidelines[6] and involves a prescription being issued by the general practitioner (GP) for patient use at a later date if symptoms do not improve or if they worsen. This can be delivered using a number of techniques, which include four common methods—providing an antibiotic prescription to the patient, dated on the day of consultation, providing a postdated prescription, the patient telephoning the practice if they meet issuing criteria for the practice to then issue a prescription, and the prescription being left at the surgery reception for the patient to collect if the patient feels it necessary. In addition, the use of self-help over-the-counter medications and techniques can also be used to help relieve symptoms of RTI without the

use of antibiotics; these include the use of paracetamol, ibuprofen and steam inhalation.[10–12]

The PIPS (Pragmatic Ibuprofen Paracetamol and Steam) trial (ISRCTN 3855126, post-results) aimed to assess the effectiveness of four common methods of delayed prescribing outlined above, and the use of analgesics (paracetamol, ibuprofen or both) and steam inhalation for RTI, using a randomised factorial trial design.[13] This paper reports on a qualitative study nested within the PIPS trial in order to determine the feasibility and acceptability of these management strategies from the perspective of participating patients and to investigate factors that are currently influencing their decisions to consult a GP.

## METHODS
### Participants and procedure
Eligible adults who had consented to participating in the main PIPS trial, and to being contacted by a researcher to discuss participating in an interview, were consecutively sampled, with ongoing monitoring for variation and prioritising the selection of individuals who would enhance the diversity of our sample where possible. This included a maximum possible variety of trial stage (from beginning to end of trial), trial arm (eg, delayed prescribing method), age and gender, to ensure we captured any potential variation in views. Participants had been recruited to the main trial through primary care by either GPs or nurses, and were randomised to 1 of 12 basic groups. Participants were recruited for this interview study by telephone from areas across Hampshire, a minimum of 2 weeks after recruitment, and written consent was obtained.

### Interviews
Trained female interviewers (LMcD, AN, JW) conducted face-to-face (n=10) and telephone interviews (in order to cover a wide geographic area, n=10), with each lasting approximately half an hour. All interviews took place at the GP clinic or the participant's home, and were audio-recorded and transcribed verbatim. Our epistemological position is best characterised by subtle realism.[14] Qualitative interviews provided the best method for gathering insights into participants' individual views about and experiences of treatments for RTI. A semistructured interview guide (online supplementary appendix 1) included key topic areas while also providing flexibility to explore unanticipated issues. Participants were asked about their experiences of RTIs to date, not solely within the context of the trial. They were also asked about different management strategies, their understanding of antibiotics and factors currently influencing their decision to consult a GP. AN and JW were medical students at the time of data collection and supervised by senior researchers (GML and PL). Everyone who consented to participate was interviewed.

### Analysis
Inductive thematic analysis[15] was conducted by hand on all transcripts to determine factors that influence patients' decision to consult a GP or use an alternative treatment for RTI, as well as being open to themes outside of the core aims of the study. Following immersion in the transcripts, familiarisation was achieved and patterns and prominent themes that consistently occurred in the data were identified and labelled with codes. Each code label referred directly to the operationalisation of the theme content. A label and full descriptive definition were then provided for each theme. Codes and definitions were refined during a continuing process, which involved themes being linked, grouped, moved, relabelled, added and removed to produce a set of themes and subthemes and a coding manual, which adequately fitted and thoroughly explained the data. The coding was iteratively developed across authors (led by LMcD and GL) and adjustments made where appropriate based on discussion. The analysis process showed that saturation had been reached as no new information was emerging from the later transcripts.

### Findings
#### Participants
Twenty people participated, with ages ranging from 18 to 74 (mean age 51), and 70% (n=14) were women. In addition, approximately 50% (n=11) of participants had completed the trial at the time of interview. All patients had experienced at least two or three different treatment options for RTI either as part of the trial or in the past.

#### Themes
Thematic analysis identified a total of five key themes relating to patients' views of different management approaches, and factors that may influence a patient's decision to use a self-help treatment (paracetamol/ibuprofen/steam inhalation) or consult with a GP for the treatment of a RTI. These are shown in table 1.

In the following sections, we summarise themes 1 and 2 for context before reporting in depth on themes 3–5, describing each in turn, using exemplary quotations for illustrative purposes (participant number is shown in parentheses). Findings are discussed in relation to the most prevalent and influential themes outlined by participants.

#### Themes 1 and 2: perceptions of illness severity, and advice from others
Participants' views on the severity of their condition related to the evaluation of the duration of symptoms. Participants signalled that they were mostly unaware of the natural history, and specific symptoms (such as breathing difficulty) were often highlighted as indicators that their condition might be severe, resulting in the decision to consult.

> I think it was because it was—I could feel or could hear a creaking or croaking in the chest cavity… So

**Table 1** Themes identified in analysis

| Themes | Subthemes | |
|---|---|---|
| 1. Perceptions of illness | | ▶ Duration of symptoms<br>▶ Perceived signs of severity<br>▶ Quality of life impact |
| 2. Advice from others | | ▶ Acceptance of GP advice<br>▶ Consideration of alternative advice |
| 3. Perceptions of antibiotics | | ▶ Antibiotics have unpleasant side effects<br>▶ Concerns and misunderstandings about resistance<br>▶ Antibiotics needed for specific cases |
| 4. Perceptions of self-help treatment | | ▶ Treatment duration short and irregular<br>▶ Experience of effective treatment |
| | Paracetamol and ibuprofen | ▶ Apprehension of combination medication<br>▶ General dislike of medication<br>▶ Concerns of medication side effects<br>▶ Medication acceptable for pain relief |
| | Steam inhalation as a limited technique | ▶ Only beneficial if not severe<br>▶ Provides short-term relief<br>▶ Relieves but not cures |
| 5. Perceptions of delayed prescribing | | ▶ Viewed as a positive option<br>▶ Acceptance of GP-recommended method<br>▶ Confusion of delayed prescribing role |

GP, general practitioner.

that said to me, right we need to go and get something to deal with it. (PL7)

Participants reported making decisions on how to treat their condition based on advice from both GPs and alternative professional and lay sources, which included pharmacists, family members and the media.

I do the Karvol inhalation, my patent remedy… That's an old one from my mother, so it's a long, long time ago, but… I think that'd be the best treatment. (P03)

### Theme 3: perceptions of antibiotics

A patient's decision on how to treat their condition was reported as being influenced by their perceptions of antibiotics, such as relating to beliefs about unpleasant side effects and concerns about resistance. Patients who described unpleasant side effects associated with the use of antibiotics tended to report reluctance to take them, unless absolutely necessary.

I don't like taking antibiotics because, after all, whatever it does to the system in relation to the bowel side of it, it destroys all the bacteria there anyhow so it's better to, if you can get away without taking them, the better. (PL08)

In line with existing literature that identified people may believe the body to be resistant rather than the bacteria,[16] many participants reported concerns about antibiotic resistance, which influenced their decision on how to best treat their condition and symptoms. Participants reported the belief that antibiotics were necessary

in some specific situations and circumstances, such as patients with comorbidities.

Well in my own particular case… I almost feel that if I do go the doctor's with a nasty cough or some sort of… you know, nasal blockage or ear ache, I almost thought I ought to be given antibiotics as a matter of course because of my asthma. (PL7)

### Theme 4: perceptions of self-help treatment

Participants' perceptions surrounding the potential benefits and limitations of self-help treatments for RTI influenced their decision to try these or visit their GP. They reported that in general, self-help treatments were used for short periods of time at irregular intervals, in contrast with the likely prescribed management approach offered by a GP to take pain relief at regular intervals.

My GP did say that managed pain is better, so take it regularly, but I tend not to do that; perhaps have some in the morning and if I feel really ropey in the afternoon, but then definitely in the evening. (PL8)

In general, participants reported a dislike for self-help treatments that involved taking medication. However, it is interesting to note that despite views of disliking these medications, participants were more likely to use them if the GP advised it and would be less likely to try these without such advice. Interviewees reported concerns and worries relating to possible side effects that taking paracetamol or ibuprofen may cause.

Yes, isn't there a problem with that (ibuprofen) in relation to blood? Yes, there's something I've heard about ibuprofen, that particular medication, which I didn't think was too good… I would not want to take it in that sense. (PL01)

Participants reported being apprehensive about taking a combination of ibuprofen and paracetamol together. The combination of medication was construed by some as somehow 'risky' when compared with taking a single type of medication.

Well I should have thought that was a bit of a lethal cocktail together but I don't know; if it was up to me I wouldn't want to do that. (PL08)

Despite concerns relating to taking paracetamol or ibuprofen for the treatment of RTI, participants reported that they were happy to take these for the use of pain relief. This was a familiar usage and appeared to be considered as more acceptable than using them for an RTI.

Most reported steam inhalation could relieve symptoms to a degree; however, it was perceived as only providing short-term relief from symptoms and that it would not assist in curing the underlying condition. Therefore, if symptoms were considered as serious, using steam inhalation was not viewed as a potential overall treatment option.

I don't think that they actually cure me, but I think they give me a little bit of relief from congestion. (PL09)

### Theme 5: perceptions of delayed prescribing
Delayed prescribing was generally viewed as a positive technique and patients reported being happy to accept whichever method of delayed prescribing the GP recommended. None of the participants expressed disappointment at not receiving an antibiotic prescription for immediate use. Delayed prescribing was generally viewed by participants as a positive technique, which could give them reassurance of having access to a prescription just in case. The main benefit was seen as the patient having some decision in their own treatment while preventing them from unnecessarily taking antibiotics.

I don't mind delayed prescribing. I mean, I would rather have them and know that I've got them and, yes, if I don't need them, then I wouldn't take them, but I like the safety (of a delayed prescription), knowing that I've got them, because more often than not, it doesn't go away. (P05)

Although happy to accept delayed prescribing as a treatment option, many participants were confused as to the purpose of the technique. Some reported confusion because they had been told their condition was viral yet had still been issued a delayed prescription, indicating the rationale for delayed prescribing may need to be clearer.

…either you have or haven't got an infection that either does or doesn't need treating. And if it needs treating it needs treating now, if not it doesn't. (PA7)

Some expressed the view that the technique might be driven by economic reasons.

I am more and more wondering if medication is being withheld because of expenses… surgeries have to make a living, they have to work within a budget… (PA3)

## DISCUSSION
### Main findings
Overall, the findings suggest participants were aware of the limitations of antibiotics and do not wish to receive them every time they consult a GP for RTIs. There was some uncertainty about the rationale for delayed prescribing with a particular risk of mixed messages, whereby on the one hand some expressed the conflict between being told their condition did not warrant immediate treatment because it may not be a bacterial infection, while on the other hand it might at some point require antibiotic treatment. This is a complicated message for GPs to deliver and for patients to receive, interpret and understand. It may be that GPs should not discuss viruses during the consultation and rather emphasise that most types of RTIs settle on their own without the need to take antibiotics. Some participants suspected that delayed prescribing was primarily driven by economic decisions and the need to cut spending, but most participants viewed delayed prescriptions positively, suggesting they were happy to accept their GP's recommendation. That said, what cannot be emphasised enough is the clear need for GPs and their patients to negotiate the delayed disposal and to render transparent the rationale for a delay.

Participants expressed concerns over self-help that involved taking medication, particularly painkillers in combination for the treatment of RTI. When patients did take medication, some indicated irregular use of them, which contrasts with a need for regular and continued dosing, sometimes when symptoms appear to have reduced. Just taking 'drugs' when they feel they need to is intuitively logical and may signal a need for clearer information that people could get better and feel better more quickly if they take medication as recommended. Indeed, it seemed overall, despite reservations, participants conveyed an overall stance of a willingness to accept advice to use painkillers if recommended by their GP. Some interviewees seemed to require authorisation from their GP to try over-the-counter remedies: this may link to a misconception that analgesics do not help for respiratory symptoms. While steam inhalation was viewed as an acceptable treatment, participants generally described this method as providing short-term relief from non-severe symptoms. The decision to consult a GP was initially influenced by patient perceptions of the natural history

and duration, a variety of personal indicators, and beliefs that contrast with documented natural history indices of severity.

## Comparison with existing literature

Participants reported being aware of the limitations of antibiotics and not expecting to receive them every time they went to the GP with a RTI, which supports previous research, and often contrasts with GP perceptions of high levels of patient pressure to prescribe.[8] However, the self-selected nature of the sample interviewed is likely to mean they were more open to exploring alternatives to antibiotics.[17] There is still considerable room for patients to have a better understanding of the limited benefit from antibiotics, such as patient information leaflets, given the evidence that beliefs in the effectiveness of antibiotics are still strong.[13 18] There is also room to ensure potential misunderstandings are managed well when a no-prescribing disposal is selected. For example, while not common, some interview accounts suggested that not prescribing antibiotics can be understood (at least in part) to be a resource-saving strategy rather than a clinically driven one. Tonkin-Crine et al[19] also found that older adults may hold such a view. In addition, the perception of a delayed script as potentially contradictory to GP advice that an antibiotic prescription may not be required also needs to be tackled when a prescribing decision is negotiated. Patients reported high levels of concordance with GP advice across all treatment options, including all methods of delayed prescribing. Previous studies suggest high patient compliance/concordance with GP advice is strongly associated with doctor–patient agreement, which is facilitated by patient understanding of treatment benefits.[20] Data presented here suggested participants would have been happy to accept a delayed prescription: these perceptions were supported by the quantitative satisfaction data from the trial.[13] Previous research identified patient satisfaction with delayed prescribing,[21] but also a subgroup of patients who had expected antibiotics and were disappointed when they did not receive a prescription for immediate use. The current study in contrast did not find a group of patients who were disappointed at not receiving antibiotics. Speculatively, this may, in part, be associated with societal changes and the extensive publicity about antibiotic resistance. However, this also may reflect selection bias since trial patients may represent a subgroup who are interested in research, have more awareness of issues surrounding antibiotic prescribing and are more open to trying different approaches and therefore happier to accept new advice or techniques. Furthermore, since all participants were taking part in a trial, it is likely that GPs provided a detailed explanation of their treatment decision, using structured advice sheets, which is unlikely to happen in routine practice.[22]

## Implications for clinical practice

Overall, participants reported delayed prescribing to be an acceptable treatment option. Previous research[23] has suggested that patient understanding of the rationale for a particular treatment can increase patient satisfaction. In this study, delayed prescribing was delivered using structured advice sheets detailing the expected natural history, benefits/disbenefits of antibiotics, how long to delay and safety netting/symptom advice. The findings suggest good concordance and high satisfaction should accompany a consistent and thorough explanation of the approach. Participants did not display a strong preference of any particular method of delayed prescribing, which suggests any approach is likely to be acceptable in practice, as long as a clear explanation is provided.

Findings suggest participants were concerned about using painkillers as a treatment method for RTI. They reported concerns relating to potential hazards of combining analgesics, which links with existing evidence of low analgesic concordance rates in patients with RTI.[24] However, interviewees did report that they thought paracetamol and ibuprofen were, at least in isolation, acceptable treatments for pain relief. The results of the main trial suggest that advice to consume regular ibuprofen either alone or in combination may be associated with worse outcomes and therefore a preference for paracetamol would be the recommended approach in future.[13] This finding suggests that some analgesics can be viewed as feasible treatment options for patients, but education and explanation of the justification and possible benefits/disbenefits for these medications need to be shared with patients, as well as emphasising the role of pharmacists in terms of offering information and support. Regarding steam inhalation, results from the PIPS trial[13] showed no impact on symptoms and minor scalds in some individuals; combined with the findings presented in this paper (mostly short-term help at best) and prior literature,[11] this means that until more robust data are available, advice to use steam cannot be justified. More frequent/prolonged use of steam might help, but this would need to be shown before a recommendation could be made.

This qualitative study shows how patients' decisions to consult a GP were often related to perceived indicators of illness severity. These findings are consistent with Leventhal et al's (1984) model of illness representations,[25] which suggests that patients' beliefs relating to the cause, identity, duration and controllability of an illness are all likely to influence the way in which they respond to the condition (eg, consulting a GP). The Genomics to combat Resistance against Antibiotics for Community acquired lower respiratory tract infection (LRTI) in Europe (GRACE) INternet Training for Reducing antibiOtic use (INTRO) study used this model to develop a patient booklet, and findings indicated it was well received by patients and gave them new information about illness duration and how to self-care.[19 26] Other resources have also been developed to improve patient understanding of RTIs in primary care.[27–29] As most of the indicators that patients used to determine illness severity may not warrant a GP consultation, it suggests a need to review the clarity of patient information (eg, providing clear examples of symptoms

and illness duration that would ordinarily merit a visit to the GP, and those that are not a cause for worry). The promotion of this information within the community as well as dealing with the common misconception that antibiotic medication may help coughs and colds[17] could help to reduce unnecessary GP consultations in the future.

### Strengths and limitations

One of the key strengths of the study is that all patients interviewed had experienced at least two or three different treatment options for RTI either as part of the trial or in the past. This included antibiotics, delayed prescribing, paracetamol, ibuprofen and steam inhalation. Therefore, the interviews provided opportunity for discussion of the various treatment options with a targeted information-rich sample. This strengthens the findings and allows stronger conclusions to be drawn.

However, the sample of participants who took part in the interviews may have limited the scope of the findings somewhat, as they had all participated in the PIPS trial. This may have led to a sample of participants who were particularly interested in research of this nature and may not have represented views held by 'typical' patients, particularly views on antibiotic resistance. Also, the participants had all consulted for an RTI and therefore may have different views than individuals who may not consult for such symptoms. While having incomplete information on patient characteristics including information on people who declined participation is a limitation of this study, the research does provide novel and relevant findings.

### CONCLUSION

The findings suggest that delayed prescribing appears to be an acceptable technique, as long as the method of delayed prescribing is described well to optimise patient understanding of the rationale. Increased patient education relating to the safety and justification of using paracetamol for the treatment of RTI is likely to increase the acceptability of analgesics. In addition, enhanced approaches to sharing information with patients—relating to the natural history of illness, the appropriate signs and symptoms requiring GP advice, and those symptoms and signs that do not require urgent attention—could help reduce unnecessary consultations.

**Acknowledgements** We are grateful to all the general practitioners and practice nurses and particularly the patients who gave their time to this study.

**Collaborators** University of Southampton: Paul Little, Ian Williamson, Mike Moore, Mark Mullee, Jo Kelly, Julie Hooper, Lisa McDermott, Geraldine Leydon, Ben Holdstock-Brown (medical student at the time), Amanda Nagle (medical student at the time), Jennifer White (medical student at the time); Samantha Hall Patient and Public (PPI) representative.

**Contributors** GML led the qualitative work as part of the PIPs trial. PL secured research funding and acted as overall PI of the PIPs trial. JK facilitated recruitment. LMcD, AN and JW collected interview data. All authors were involved in/commented on data analysis (led by GML and LMcD). AH and GML developed the manuscript, to which all authors contributed.

**Funding** This article presents independent research funded by the National Institute for Health Research (NIHR) under its Programme Grants for Applied Research programme (grant ref no. RP-PG-0407-10098). Thanks to NIHR Post Doctoral Fellowship for funding GML during this work (grant ref no. NIHR/PDF-2009-02-35).

**Disclaimer** The views expressed are those of the authors and not necessarily those of the NHS, the NIHR or the Department of Health.

**Competing interests** None declared.

**Ethics approval** The study was approved by Southampton and Southwest MREC number 06/Q1702/154.

**Provenance and peer review** Not commissioned; externally peer reviewed.

**Data sharing statement** No additional data available.

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
