## [Reviewer comments · BMJ Open]

ARTICLE DETAILS

TITLE (PROVISIONAL)	A qualitative interview study of antibiotics and self-management strategies for respiratory infections in primary care.
AUTHORS	McDermott, Lisa; Leydon, Gerry; Halls, Amy; Kelly, Jo; Nagle, Amanda; White, Jennifer; Little, Paul.

VERSION 1 - REVIEW

REVIEWER	Clodna McNulty Public Health England Primary Care Unit England Our team is responsible for PHE National antibiotic quick reference guide and the TARGET antibiotics toolkit that encourages delayed prescribing
REVIEW RETURNED	12-Apr-2017

GENERAL COMMENTS	1) Page 2 Line 49 Point 1 – Later in the paper you state information collected about delayed prescribing was limited – so is it better to put wording as in objectives.2) Page 2 Line 53 Point 2 – Add data saturation.3) Page 3 Line 11 Background – Surely antibiotics do not account for 60% of all prescriptions in primary care. Do you mean that RTIs are the reason for 60% of all antibiotic prescribing in primary care?4) Page 3 Line 18 – Definition of delayed antibiotic is broader please amend, as also used if symptoms get worse.5) Page 3 Line 21 – Unaltered – may be better as provide an antibiotic prescription in the usual way, dated on day of consultation.6) Page 3 Line 21 – Make clear that the patient phones the practice if meet issuing criteria and the practices then issues it.7) Page 3 Line 25 – Make sure that ‘they’ is understood as the patient not reception.8) Page 4 Line 37 – A short last sentence in the analysis saying what you did with codes, or how you arrived at the final five key themes, and who discussed this.9) Page 6 Line 46-47 – This is more for the discussion, but dose
--

	depend on BMJ open house style. 10) Page 9 Line 13-17 – Not necessary. 11) Page 10 Line 23 - ‘better understanding’ – clarify if this is patients and what information should be given. 12) Page 10 Line 25-32 – Rewrite to make clearer – 2 sentences seem disconnected. 13) Page 10 Line 35 – Explain contradictory in this sentence – what should the prescriber say? 14) Page 10 Line 40 – Compliance/concordance with what? 15) Page 10 Line 42 – ‘Although not discussed in detail’ – This is interesting as the title of your paper suggests that it would have been. Therefore I suggest you remove this from the title. 16) Page 11 Line 27-28 – I suggest a simplified version or bullet points of the GP information is provided as a box/table/figure in the paper if you are suggesting this is used. 17) Page 11 Lines 32-47 – Mixed messages here – actually you are suggesting that ibuprofen should not be used – so maybe the patients were correct to be concerned. Need clearer approach and discuss role of pharmacist in giving useful advice or need to put on leaflets regular paracetamol. 18) Page 12 Line 8 – Would be useful to also quote Francis et al When should I worry leaflet, Hay et al Childrens cough poster and TARGET Treat your infection leaflet. 19) Add to discussion – If there was mixed messages about viruses and use of delayed antibiotics, it may be an appropriate conclusion for GPs not to discuss viruses in the consultation, and say rather that most of the types of infection settle on their own without the need for antibiotics. 20) Page 12 Line 49 – Is the word ‘disposal’ appropriate ‘disposal’ means to ‘throw away’ – Rewrite sentence to clarify meaning. 21) Page 2 Line 38 – Mixed messages – Add about severity of illnesses and causation by viruses or bacteria. 22) Page 2 Line 41 – As you explored just 3 things is it better just to put these in – paracetamol, ibuprofen and steam inhalation? Or should you actually not be promoting ibuprofen because of other trial results. 23) References – It would be useful to do a more recent check for other references on this area.
--	---

REVIEWER	Joanna Kesten University of Bristol
REVIEW RETURNED	20-Apr-2017

GENERAL COMMENTS	COREQ
-------

Please fully complete the COREQ checklist.

Title

Consider deleting “using” as “use of using” does not read very well. Personally, I do not feel the quote (lethal cocktail) is very clear out of context. Please consider removing or perhaps replacing with another quote.

Abstract

Participants: For clarity, please specify that adult patients were interviewed.

Results: Please could you clarify/expand on the “significant concerns about symptoms which were not serious”. What were patients concerned about for non-serious symptoms?

In the sentence beginning, “Most viewed delayed prescriptions positively” please replace “it” with “their infection”.

Article summary

Point 5. In addition, to the trial participants potentially being particularly interested in research is there also the possibility that the interview sample were biased as all participants had consulted for an RTI and therefore may have different attitudes and views than those who do not consult for these symptoms? This is assuming that trial participants were recruited at the time of consultation (see later comment about trial description).

Background

Is there any literature available on how clinicians choose between the four delayed prescribing techniques? Similarly, is there any evidence about which of these methods is most/least commonly used by clinicians? I think this would be useful context for the background section.

It would be useful to know more about the trial in which this study is nested. Please provide a reference to a PIP trial paper, for example a protocol paper, and briefly summarise what the trial involved. The limitations section highlights that trial participation may have influenced the findings of this qualitative study so it is pertinent for readers to know more about the trial processes. This information could be included in the background or methods section.

Methods

Participants and procedure: please state that participants were adult. Please provide more detail on how participants were recruited to the trial? For example, through which setting (e.g. GP surgery), using which method (e.g. letters, posters in surgeries) and was recruitment purposeful/random Etc.

Please explain the reasons for participants being sampled to include a “maximum possible variety of trial stage” and “trial arm”.

Please explain why telephone interviews were chosen as the method of data collection as opposed to face-to-face interviews?

It would be useful if Appendix 1 could be more detailed. It is not clear whether the interviews primarily focused on the RTI within the trial or general experiences of RTIs and treatments used ever or those during the trial period. Were all participants asked about every trial treatment option including those they were not randomised to? I.e. if randomised to delayed prescription were they asked to reflect on views of steam, antibiotics etc.?

Although the analysis is described as inductive, please can the authors reflect on the extent to which the themes somewhat

deductively reflect the interview topic guide.
Please describe how the decision to stop data collection was made. The 'Article summary' section states that "Saturation was reached", this needs to be described in more detail within the main body of the manuscript.

Please describe the recruitment rate. What proportion of those invited declined to participate?

Results

Please include the duration of the interviews.

How does the achieved sample relate to the overall PIP trial cohort? Approximately 50% of participants had completed the trial at the time of interview, at what stage of the trial were the other participants?

Table 1. In relation to self-help treatments, the themes only relate to analgesics and steam, were any other self-help treatments identified by the participants? If no other self-help treatments were described I wonder if this reflects the topic guide and whether participants were given an opportunity to identify their own self-help treatments.

It would be useful to note whether there were any differences in perceptions of the trial arms between those who had vs those who had not received them. E.g. did those who received delayed prescribing have different attitudes towards it compared to those hypothetically commenting on its acceptability?

Given the diverse sample, please consider including a more detailed description alongside each quotation to reflect the purposeful sampling approach. For example, age, gender, trial arm etc.

Alternatively, please consider providing a table of these characteristics for each participant.

Theme 1 & 2. I appreciate you are trying to be succinct in these two themes and I wonder if it would be better to remove these themes in favour of presenting more detailed findings for themes 3-5 which are more novel? If you decide not to do this, it would be clearer if the formatting of the heading for theme 1 and 2 is the same as the other themes.

Please provide more detailed findings relating to the factors influencing the source of advice or help-seeking (e.g. consulting a GP, lay sources etc.)?

Theme 4. Please provide more detail regarding how perceptions surrounding the potential benefits and limitations of self-help treatments for RTI inform decisions to consult the GP?

There is some repetition in the results. For example, "antibiotics are required for specific conditions and situations" and "antibiotics were necessary in some specific situations and circumstances". Also, the sentence beginning "The combination of medication was construed" could be combined with "Interviewees reported concerns and worries relating to possible side effects".

Did the interviews elicit any more detail on the relative merits of the different delayed prescribing methods? It is not clear whether participants were aware of the various methods or if these were explained during the interviews?

Were perceptions of the merits of self-help vs delayed prescribing discussed in the interviews at all or were these items discussed in isolation during the interviews? i.e. would either of these be more or less acceptable/preferable to patients?

Discussion

I think the second sentence "Findings are discussed" would fit better at the beginning of the results section.

	Main findings: Please insert “for RTIs” at the end of the first sentence. Comparison with existing literature: Does the following sentence relate to the qualitative findings: “The current study in contrast did not find a group of patients who were disappointed at not receiving antibiotics”. If so, please add this to the results section and consider making the acceptability of being recommended self-treatment advice by GPs rather than antibiotics more explicit. Implications for clinical practice, paragraph 3: The following finding does not appear to be presented in the results “Patients’ decisions to consult a GP were often related to perceived indicators of illness severity”. In addition, how was accuracy of illness severity (“more often incorrect than correct”) assessed in this qualitative study? Please clarify if this sentence relates to results from the main trial. Ethical approval Was informed consent for interviews received in written or verbal form?
--	---

VERSION 1 – AUTHOR RESPONSE

REVIEWER 1 COMMENTS

AUTHOR RESPONSE

1 Page 2 Line 49 Point 1 – Later in the paper you state information collected about delayed prescribing was limited – so is it better to put wording as in objectives.

We have kept this wording as we believe the interviews did allow for an open exploration, even if the findings were somewhat limited. However we have reordered this section, so this specific strength is now the third of five.

2 Page 2 Line 53 Point 2 – Add data saturation.

Thank you for this observation, we have added in ‘data’ so this strength now reads as follows: “1. A range of views were solicited and data saturation was reached.”

3 Page 3 Line 11 Background – Surely antibiotics do not account for 60% of all prescriptions in primary care. Do you mean that RTIs are the reason for 60% of all antibiotic prescribing in primary care?

Thank you for highlighting this error. We have edited it and the sentence now reads as follows on page 3: “Despite evidence to suggest the limited benefit of antibiotics in the treatment of respiratory tract infections (RTI) in particular[4-5], RTIs account for >60% of all antibiotic prescriptions in primary care[6]”

4 Page 3 Line 18 – Definition of delayed antibiotic is broader please amend, as also used if symptoms get worse.

Thank you for this clarification, we have edited the sentence which now reads as follows on page 3: “The method is recommended by the NICE guidelines[6] and involves a prescription being issued by the GP for patient use at a later date if symptoms do not improve, or if they worsen.”

Comments 5-7 have been addressed together:

5 Page 3 Line 21 – Unaltered – may be better as provide an antibiotic prescription in the usual way, dated on day of consultation.

Thank you for these three comments, this section has been edited for clarity and now reads as follows on page 3: “This can be delivered using a number of techniques which include four common methods – providing an antibiotic prescription to the patient, dated on the day of consultation, providing a post-dated prescription, the patient telephoning the practice if they meet issuing criteria for the practice to then issue a prescription, and the prescription being left at the surgery reception for the patient to

collect if the patient feels it necessary.”

6 Page 3 Line 21 – Make clear that the patient phones the practice if meet issuing criteria and the practices then issues it.

7 Page 3 Line 25 – Make sure that ‘they’ is understood as the patient not reception.

8 Page 4 Line 37 – A short last sentence in the analysis saying what you did with codes, or how you arrived at the final five key themes, and who discussed this.

We have discussed this and believe the description of analysis describes how we arrived at the final key themes (we can amend if the editorial team view this as problematic). We have added that this was done with the full team in team meetings.

9 Page 6 Line 46-47 – This is more for the discussion, but dose depend on BMJ open house style.

We think it is helpful to signpost to this reference in this section, but we are happy to move this to the Discussion following guidance from the editorial team.

10 Page 9 Line 13-17 – Not necessary.

The first sentence has been deleted as suggested, and the second sentence has been moved as suggested by Reviewer 2.

11 Page 10 Line 23 - ‘better understanding’ – clarify if this is patients and what information should be given.

Thank you for this comment, we have edited this sentence on page 11 so it is clearer: “There is still considerable room for patients to have a better understanding of the limited benefit from antibiotics, such a patient information leaflets, given the evidence that beliefs in the effectiveness of antibiotics are still strong[13,17].”

12 Page 10 Line 25-32 – Rewrite to make clearer – 2 sentences seem disconnected.

Thank you for the comment. We have edited these sentences on page 11 for clarity and they now read as follows: “There is also room to ensure potential misunderstandings are managed well when a no-prescribing disposal is selected. For example, whilst not common, some interview accounts suggested that not prescribing antibiotics can be understood (at least in part) to be a resource saving strategy rather than a clinically driven one.”

13 Page 10 Line 35 – Explain contradictory in this sentence – what should the prescriber say?

Thank you for this comment. This sentence has now been edited on page 11 and is hopefully clearer. It reads as follows: “In addition, the perception of a delayed script as potentially contradictory to GP advice that an antibiotic prescription may not be required also needs to be tackled when a prescribing decision is negotiated.”

14 Page 10 Line 40 – Compliance/concordance with what?

Thank you for this comment. The sentence has now been edited on page 11 for clarity and reads as follows: “Previous studies suggest high patient compliance/concordance with GP advice is strongly associated with doctor-patient agreement, which is facilitated by patient understanding of treatment benefits[18].”

15 Page 10 Line 42 – ‘Although not discussed in detail’ – This is interesting as the title of your paper suggests that it would have been. Therefore I suggest you remove this from the title.

Thank you for this comment, this has been edited on page 11 and now reads as follows: “Data presented here suggested participants would have been happy to accept a delayed prescription: these perceptions were supported by the quantitative satisfaction data from the trial[15].” The title has also been edited and is now: A qualitative interview study of antibiotics and self-management strategies for respiratory infections in primary care.

16 Page 11 Line 27-28 – I suggest a simplified version or bullet points of the GP information is provided as a box/table/figure in the paper if you are suggesting this is used.

Thank you for this comment. For clarity, this sentence has been removed.

17 Page 11 Lines 32-47 – Mixed messages here – actually you are suggesting that ibuprofen should not be used – so maybe the patients were correct to be concerned. Need clearer approach and discuss role of pharmacist in giving useful advice or need to put on leaflets regular paracetamol. Thank you for this comment, we agree it could be clearer. We have edited this paragraph and it now reads as follows: “Findings suggest participants were concerned about using painkillers as a treatment method for RTI. They reported concerns relating to potential hazards of combining analgesics, which links with existing evidence of low analgesic concordance rates in patients with RTI[22]. However, interviewees did report that they thought paracetamol and ibuprofen were, at least in isolation, acceptable treatments for pain relief. The results of the main trial suggest that advice to consume regular ibuprofen either alone or in combination may be associated with worse outcomes and therefore a preference for paracetamol would be the recommended approach in future[15]. This finding suggests that some analgesics can be viewed as feasible treatment options for patients, but education and explanation of the justification and possible benefits/dis-benefits for these medications need to be shared with patients, as well as emphasising the role of pharmacists in terms of offering information and support.”

18 Page 12 Line 8 – Would be useful to also quote Francis et al When should I worry leaflet, Hay et al Childrens cough poster and TARGET Treat your infection leaflet.

Thank you for this suggestion, these references have been added in and are on page 13 and are references 27-9.

19 Add to discussion – If there was mixed messages about viruses and use of delayed antibiotics, it may be an appropriate conclusion for GPs not to discuss viruses in the consultation, and say rather that most of the types of infection settle on their own without the need for antibiotics.

Thank you for this very valid discussion point. We have added it in on page 10 and it now reads as follows: “This is a complicated message for GPs to deliver and for patients to receive, interpret and understand. It may be that GPs should not discuss viruses during the consultation, and rather emphasise that most types of RTIs settle on their own without the need to antibiotics.”

20 Page 12 Line 49 – Is the word ‘disposal’ appropriate ‘disposal’ means to ‘throw away’ – Rewrite sentence to clarify meaning.

Thank you, we have edited this sentence and it now reads as follows on page 14: “The findings suggest that delayed prescribing appears to be an acceptable technique, as long as the method of delayed prescribing is described well to optimise patient understanding of the rationale.”

21 Page 2 Line 38 – Mixed messages – Add about severity of illnesses and causation by viruses or bacteria.

Thank you for this comment, we have edited the sentence following your suggestion and this is how it now reads: “Delayed prescribing is acceptable no matter how the delay is operationalized, but explanation of the rationale is needed and care taken to minimise mixed messages about the severity of illnesses and causation by viruses or bacteria.”

22 Page 2 Line 41 – As you explored just 3 things is it better just to put these in – paracetamol, ibuprofen and steam inhalation? Or should you actually not be promoting ibuprofen because of other trial results.

Thank you for this observation, we have edited the sentence in question and it now reads as follows on page 2: “Significant concerns about paracetamol, ibuprofen and steam inhalation are likely to need careful exploration in the consultation.”

23 References – It would be useful to do a more recent check for other references on this area.

Thank you for this suggestion, we have updated the references list, for example De la Poza Abad MD et al: Prescription strategies in acute uncomplicated respiratory infections: A randomized clinical trial. JAMA Intern Med. 2016;176(1):21-29. doi:10.1001/jamainternmed.2015.7088

REVIEWER 2 COMMENTS

AUTHOR RESPONSE

1 The main strength and novelty of this study is the exploration of attitudes towards self-management and delayed prescribing techniques for the management of RTIs. Thank you. We are pleased that you find our work to be novel.

2 Please fully complete the COREQ checklist

We do apologise that the COREQ checklist originally submitted was incomplete. Please find a completed version attached as part of this resubmission.

Comments 3-4 have been addressed together

3 Consider deleting “using” as “use of using” does not read very well. Thank you, this has been done. The title is now: ‘A qualitative interview study of antibiotics and self-management strategies for respiratory infections in primary care’.

4 Personally, I do not feel the quote (lethal cocktail) is very clear out of context. Please consider removing or perhaps replacing with another quote.

5 Abstract

Participants: For clarity, please specify that adult patients were interviewed.

Thank you for this observation, this has been edited and now reads as follows: “Participants: 20 adult patients who had been participating in the ‘PIPS’ (Pragmatic Ibuprofen Paracetamol and Steam) trial in the South of England.”

6 Please could you clarify/expand on the “significant concerns about symptoms which were not serious”. What were patients concerned about for non-serious symptoms? We have edited this sentence and hope it is clearer. It reads as follows: “Significant concerns about paracetamol, ibuprofen and steam inhalation are likely to need careful exploration in the consultation.”

7 In the sentence beginning, “Most viewed delayed prescriptions positively” please replace “it” with “their infection”. Thank you, it has been edited as requested.

8 Point 5. In addition, to the trial participants potentially being particularly interested in research is there also the possibility that the interview sample were biased as all participants had consulted for an RTI and therefore may have different attitudes and views than those who do not consult for these symptoms? This is assuming that trial participants were recruited at the time of consultation (see later comment about trial description).

Thank you for this comment, we agree that this is a limitation and have added a sentence at the end of the Strengths and Limitations section. It reads as follows: “The participants had all consulted for an RTI and therefore may have different views than individuals who may not consult for such symptoms.”

9 Is there any literature available on how clinicians choose between the four delayed prescribing techniques? Similarly, is there any evidence about which of these methods is most/least commonly used by clinicians? I think this would be useful context for the background section. We are unaware of any literature on this area. These methods were chosen in conjunction with colleagues who use them. Crucially, the trial did not show much difference between them if properly used. We are happy to add in any references if the editorial team is able to suggest any.

10 It would be useful to know more about the trial in which this study is nested. Please provide a reference to a PIP trial paper, for example a protocol paper, and briefly summarise what the trial

involved. The limitations section highlights that trial participation may have influenced the findings of this qualitative study so it is pertinent for readers to know more about the trial processes. This information could be included in the background or methods section. Thank you for this comment. The PIPS trial paper was originally reference 15. In this revised document it is reference 13 and is in the Background section, as follows: "The PIPS (Pragmatic Ibuprofen Paracetamol and Steam) trial aimed to assess the effectiveness of four common methods of delayed prescribing outlined above, and the use of analgesics (paracetamol; ibuprofen or both) and steam inhalation for RTI, using a randomised factorial trial design[13]."

11 Methods

Participants and procedure: please state that participants were adult.
Thank you, this has been edited as requested.

Comments 12-13 have been addressed together:

12 Please provide more detail on how participants were recruited to the trial? For example, through which setting (e.g. GP surgery), using which method (e.g. letters, posters in surgeries) and was recruitment purposeful/random Etc.

This research project involved BMed Sci Medical Students, who graduated before overall project completion. Unfortunately, this, combined with staff leaving, resulted in some loss of procedural information. However, we believe that the findings are of interest and relevant to the ongoing scientific accumulation of knowledge surrounding the use of antibiotics and self-management strategies within primary care.

The team are experienced and conducted the study in a rigorous fashion.

13 Please explain the reasons for participants being sampled to include a "maximum possible variety of trial stage" and "trial arm".

14 Please explain why telephone interviews were chosen as the method of data collection as opposed to face-to-face interviews? Thank you for this comment, we have edited this section to aid clarity. It now reads as follows: "Trained female interviewers (LM, AN, JW) conducted face to face (n=10) and telephone interviews (in order to cover a wide geographic area, n=10), with each lasting approximately half an hour. All interviews were audio-recorded and transcribed verbatim."

15 It would be useful if Appendix 1 could be more detailed. It is not clear whether the interviews primarily focused on the RTI within the trial or general experiences of RTIs and treatments used ever or those during the trial period. Were all participants asked about every trial treatment option including those they were not randomised to? I.e. if randomised to delayed prescription were they asked to reflect on views of steam, antibiotics etc.? Thank you for this observation, we apologise as the full interview guide should have been included in the original submission. The full interview guide is now included as Appendix 1. Participants were asked about their experiences of RTIs to date, not solely within the context of the trial. They were also asked about different management strategies, their understanding of antibiotics, and factors currently influencing their decision to consult a GP.

16 Although the analysis is described as inductive, please can the authors reflect on the extent to which the themes somewhat deductively reflect the interview topic guide.

Thank you for this comment, we agree this may suggest a more deductive approach. We were open to themes outside of the core aims of the study and it was an inductive process. We have edited the first sentence in the Analysis section on page x which now reads as follows: "Inductive thematic analysis[13] was conducted on all transcripts to determine factors which influence patients' decision to consult a GP or use an alternative treatment for RTI, as well as being open to themes outside of the core aims of the study."

17 Please describe how the decision to stop data collection was made. The 'Article summary' section states that "Saturation was reached", this needs to be described in more detail within the main body of the manuscript.

Thank you for this comment, we have edited the Interviews and Analysis sections so hopefully this is clearer. Everyone who consented to participate in this stage of the research was interviewed, and the analysis process showed that no new themes were emerging from the later interview transcripts. "The analysis process showed that saturation had been reached as no new information was emerging from the later transcripts."

18 Please describe the recruitment rate. What proportion of those invited declined to participate? We acknowledge this is a limitation, but we do not hold this information. We are clear in our limitations section about this.

19 Results

Please include the duration of the interviews.

Thank you for this comment. This is partly addressed in the Methods section. "Trained female interviewers (LM, AN, JW) conducted face to face (n=10) and telephone interviews (in order to cover a wide geographic area, n=10), with each lasting approximately half an hour." Unfortunately, we do not have a record of the duration of all of our interviews. We had ethical approval for interviews to last up to 60 minutes and we know that no interview exceeded this length.

Comments 20-23 are addressed together:

20 How does the achieved sample relate to the overall PIP trial cohort?

Unfortunately, we are unable to provide further information. This was originally a research project for medical students, who graduated before overall project completion. This, combined with staff turnover, has resulted in some information missing. However, we believe that the findings are of interest and relevant to the ongoing development of knowledge surrounding the use of antibiotics and self-management strategies within primary care. The team are experienced and the project execution from start to finish was rigorous and we hope the reviewer can still see the novelty and credibility of the paper in this context.

21 Approximately 50% of participants had completed the trial at the time of interview, at what stage of the trial were the other participants?

22 It would be useful to note whether there were any differences in perceptions of the trial arms between those who had vs those who had not received them. E.g. did those who received delayed prescribing have different attitudes towards it compared to those hypothetically commenting on its acceptability?

23 Given the diverse sample, please consider including a more detailed description alongside each quotation to reflect the purposeful sampling approach. For example, age, gender, trial arm etc. Alternatively, please consider providing a table of these characteristics for each participant.

24 Table 1. In relation to self-help treatments, the themes only relate to analgesics and steam, were any other self-help treatments identified by the participants? If no other self-help treatments were described I wonder if this reflects the topic guide and whether participants were given an opportunity to identify their own self-help treatments.

Thank you for this observation. No other self-help treatments were mentioned by participants. Appendix 1 now shows our detailed interview guide which we acknowledge should have been included before. It is more detailed and does show participants were asked an open-ended question about any other self-help measures they may have used.

25 Theme 1 & 2. I appreciate you are trying to be succinct in these two themes and I wonder if it would be better to remove these themes in favour of presenting more detailed findings for themes 3-5 which are more novel? If you decide not to do this, it would be clearer if the formatting of the heading for theme 1 and 2 is the same as the other themes. Thank you for this comment. Other reviewers

were not indicating a need to remove these themes and we agree. We have changed the formatting so that there is consistency throughout the reporting of the themes.

26 Please provide more detailed findings relating to the factors influencing the source of advice or help-seeking (e.g. consulting a GP, lay sources etc.)?

We believe our report of the findings captures the key factors and have reported all key findings.

27 Theme 4. Please provide more detail regarding how perceptions surrounding the potential benefits and limitations of self-help treatments for RTI inform decisions to consult the GP?

There is some repetition in the results. For example, “antibiotics are required for specific conditions and situations” and “antibiotics were necessary in some specific situations and circumstances”. Also, the sentence beginning “The combination of medication was construed” could be combined with “Interviewees reported concerns and worries relating to possible side effects”. Unfortunately we do not have further information. We believe this theme accurately reflects the research findings and adequately conveys the theme content.

28 There is some repetition in the results. For example, “antibiotics are required for specific conditions and situations” and “antibiotics were necessary in some specific situations and circumstances”.

Thank you for this observation. This section has been edited to avoid repetition. It now reads as follows: “A patient’s decision on how to treat their condition was reported as being influenced by their perceptions of antibiotics, such as relating to beliefs about unpleasant side effects and concerns about resistance. Patients who described unpleasant side-effects associated with the use of antibiotics tended to report reluctance to take them, unless absolutely necessary.

“I don’t like taking antibiotics because, after all, whatever it does to the system in relation to the bowel side of it, it destroys all the bacteria there anyhow so it’s better to, if you can get away without taking them, the better”(PL08)

In line with existing literature that identified people may believe the body to be resistant rather than the bacteria[14], many participants reported concerns about antibiotic resistance, which influenced their decision on how to best treat their condition and symptoms. Participants reported the belief that antibiotics were necessary in some specific situations and circumstances, such as patients with co-morbidities.”

29 Also, the sentence beginning “The combination of medication was construed” could be combined with “Interviewees reported concerns and worries relating to possible side effects”.

Thank you for this comment. These two sentences and illustrative quotes have been reordered and it now reads as follows: “Interviewees reported concerns and worries relating to possible side effects which taking paracetamol or ibuprofen may cause.

“Yes, isn’t there a problem with that [ibuprofen] in relation to blood? Yes, there’s something I’ve heard about ibuprofen, that particular medication, which I didn’t think was too good...I would not want to take it in that sense.”(PL01)

Participants reported being apprehensive about taking a combination of ibuprofen and paracetamol together. The combination of medication was construed by some as somehow ‘risky’ when compared to taking a single type of medication.

“Well I should have thought that was a bit of a lethal cocktail together but I don’t know; if it was up to me I wouldn’t want to do that.”(PL08)”

30 Did the interviews elicit any more detail on the relative merits of the different delayed prescribing methods? It is not clear whether participants were aware of the various methods or if these were explained during the interviews? Thank you for this comment. Interviewers did not explore patient

preference or differential understanding of the different methods, it was an additional exploration of people's understanding of the different methods.

31 Were perceptions of the merits of self-help vs delayed prescribing discussed in the interviews at all or were these items discussed in isolation during the interviews? i.e. would either of these be more or less acceptable/preferable to patients?

Thank you for this comment. Self-help measures and delayed prescribing were addressed separately during the interviews and we do not have enough information to be able to say whether one approach would be seen as more or less acceptable/preferable to patients.

32 Discussion

I think the second sentence "Findings are discussed" would fit better at the beginning of the results section. Thank you for this comment. This has been moved and is now after the themes table on page 6.

33 Main findings: Please insert "for RTIs" at the end of the first sentence.

Thank you for this observation, "for RTIs has been added in as requested.

34 Comparison with existing literature: Does the following sentence relate to the qualitative findings: "The current study in contrast did not find a group of patients who were disappointed at not receiving antibiotics". If so, please add this to the results section and consider making the acceptability of being recommended self-treatment advice by GPs rather than antibiotics more explicit.

Thank you for these comments. We agree this should be in the Results section and theme 5 has been edited as follows on page 9: "Delayed prescribing was generally viewed as a positive technique and patients reported being happy to accept whichever method of delayed prescribing the GP recommended. None of the participants expressed disappointment at not receiving an antibiotic prescription for immediate use."

35 Implications for clinical practice, paragraph 3: The following finding does not appear to be presented in the results "Patients' decisions to consult a GP were often related to perceived indicators of illness severity". In addition, how was accuracy of illness severity ("more often incorrect than correct") assessed in this qualitative study? Please clarify if this sentence relates to results from the main trial.

This refers to theme 1 on page 6. "Participants signalled that they were mostly unaware of the natural history, and specific symptoms (such as breathing difficulty) were often highlighted as indicators that their condition might be severe, resulting in the decision to consult." Patients themselves felt that their symptoms were severe enough to consult, and a focus was often breathing and associated difficulties. Accuracy of severity was not assessed as such, the focus was on participants' own experiences and perceptions of severity.

For clarity, we have edited the first sentence in this paragraph, which now reads as follows: "This study shows how patients' decisions to consult a GP were often related to perceived indicators of illness severity which were more often incorrect than correct, as well as beliefs relating to whether they thought they had caught the same condition as other people they knew."

36 Ethical approval

Was informed consent for interviews received in written or verbal form?

Thank you for this observation, we apologise as this should have been included. It has been added to the Participants and Procedure section and now reads as follows: "Participants were recruited by telephone and written consent was obtained."

VERSION 2 – REVIEW

REVIEWER	Joanna Kesten University of Bristol, United Kingdom
REVIEW RETURNED	26-Jun-2017

GENERAL COMMENTS	“12 Please provide more detail on how participants were recruited to the trial? For example, through which setting (e.g. GP surgery), using which method (e.g. letters, posters in surgeries) and was recruitment purposeful/random Etc.” Thank you for your honest response regarding the loss of procedural information. For transparency, I suggest you describe where possible the procedure outlined in the protocol approved by the ethics committee and then note that in practice this may have been different due to loss of procedural information. Please briefly state that participants were recruited to the main trial through primary care by healthcare professionals (GPs or nurses etc.). This information is stated in ref. 13. “13. Please explain the reasons for participants being sampled to include a “maximum possible variety of trial stage” and “trial arm”.” Please provide a justification for this. The loss of procedural information should not affect this reporting. “15. It would be useful if Appendix 1 could be more detailed. It is not clear whether the interviews primarily focused on the RTI within the trial or general experiences of RTIs and treatments used ever or those during the trial period. Were all participants asked about every trial treatment option including those they were not randomised to? I.e. if randomised to delayed prescription were they asked to reflect on views of steam, antibiotics etc.?” Thank you for this observation, we apologise as the full interview guide should have been included in the original submission. The full interview guide is now included as Appendix 1. Participants were asked about their experiences of RTIs to date, not solely within the context of the trial. They were also asked about different management strategies, their understanding of antibiotics, and factors currently influencing their decision to consult a GP.” Thank you for providing the interview topic guide as an appendix. For transparency, please include the information provided above “Participants were asked about their experiences of RTIs to date... etc.” in the manuscript. Appendix There do not appear to be any questions relating to steam inhalation for RTI in the topic guide. Is this an omission? Please clarify. Analysis Please state in the manuscript whether the analysis was conducted using any software. “28 There is some repetition in the results.”
---

	Thank you for making the changes outlined. In addition, line 13 “However, there was some confusion reported as to the role of delayed prescribing” is repeated on line 24. I suggest deleting line 13. Discussion Line 7-8, page 13. The finding that “beliefs relating to whether they thought they had caught the same condition as other people they knew” is not presented in the results. Please either add to the results or remove from here. The use of steam inhalation is not discussed in detail in this section. What conclusions do the authors draw in relation to implications for clinical practice for this treatment option in relation to the literature? “35 Implications for clinical practice, paragraph 3: how was accuracy of illness severity (“more often incorrect than correct”) assessed in this qualitative study? Please clarify if this sentence relates to results from the main trial. If the focus is on valuing participants’ own perceptions of severity, I am unclear how their perceptions of the severity of their symptoms can be deemed “more often incorrect than correct”. Please clarify how this judgement has been made?
--	---

VERSION 2 – AUTHOR RESPONSE

Response to Reviewer Comments – second resubmission

Comment:

Thank you for your thoughtful and considered response to my comments. I have a small number of outstanding queries relating to the transparency of the reporting. These comments are detailed below beneath the relevant comment.

Response:

We are pleased that you found our responses to be thoughtful and considered. We have addressed your outstanding queries below.

Comment:

“12 Please provide more detail on how participants were recruited to the trial? For example, through which setting (e.g. GP surgery), using which method (e.g. letters, posters in surgeries) and was recruitment purposeful/random Etc.”

Thank you for your honest response regarding the loss of procedural information. For transparency, I suggest you describe where possible the procedure outlined in the protocol approved by the ethics committee and then note that in practice this may have been different due to loss of procedural information.

Response:

We are pleased our honest response was appreciated, and we have been able to retrieve previously unavailable information about recruitment. This has been added to the manuscript and our 'Participants and procedure' section on pages 4-5 now reads as follows: "Eligible adults who had consented to participating in the main PIPS trial, and to being contacted by a researcher to discuss participating in an interview, were consecutively sampled, with on-going monitoring for variation and prioritising the selection of individuals who would enhance the diversity of our sample where possible. This included a maximum possible variety of trial stage (from beginning to end of trial), trial arm (e.g. delayed prescribing method), age and gender, to ensure we captured any potential variation in views. Participants had been recruited to the main trial through primary care by either GPs or nurses, and were randomised to one of twelve basic groups. Participants were recruited for this interview study by telephone from areas across Hampshire, a minimum of two weeks after recruitment and written consent was obtained."

Comment:

Please briefly state that participants were recruited to the main trial through primary care by healthcare professionals (GPs or nurses etc.). This information is stated in ref. 13.

Response:

Thank you for this suggestion, we have added a sentence accordingly. It now reads as follows on page 4: "Participants had been recruited to the main trial through primary care by either GPs or nurses, and were randomised to one of twelve basic groups."

Comment:

"13. Please explain the reasons for participants being sampled to include a "maximum possible variety of trial stage" and "trial arm"."

Please provide a justification for this. The loss of procedural information should not affect this reporting.

Response:

We have edited the 'Participants and procedure' section, which now includes the follow on page 4: "Eligible adults who had consented to participating in the main PIPS trial, and to being contacted by a researcher to discuss participating in an interview, were consecutively sampled, with on-going monitoring for variation and prioritising the selection of individuals who would enhance the diversity of our sample where possible. This included a maximum possible variety of trial stage (from beginning to end of trial), trial arm (e.g. delayed prescribing method), age and gender, to ensure we captured any potential variation in views."

Comment:

"15. It would be useful if Appendix 1 could be more detailed. It is not clear whether the interviews primarily focused on the RTI within the trial or general experiences of RTIs and treatments used ever or those during the trial period. Were all participants asked about every trial treatment option including those they were not randomised to? I.e. if randomised to delayed prescription were they asked to

reflect on views of steam, antibiotics etc.?

Thank you for this observation, we apologise as the full interview guide should have been included in the original submission. The full interview guide is now included as Appendix 1. Participants were asked about their experiences of RTIs to date, not solely within the context of the trial. They were also asked about different management strategies, their understanding of antibiotics, and factors currently influencing their decision to consult a GP.”

Thank you for providing the interview topic guide as an appendix. For transparency, please include the information provided above “Participants were asked about their experiences of RTIs to date... etc.” in the manuscript.

Response:

Thank you for this comment, we have included these sentences as suggested. They are now in the ‘Interviews’ section on page 4, which reads as follows: Participants were asked about their experiences of RTIs to date, not solely within the context of the trial. They were also asked about different management strategies, their understanding of antibiotics, and factors currently influencing their decision to consult a GP.”

Appendix

Comment:

There do not appear to be any questions relating to steam inhalation for RTI in the topic guide. Is this an omission? Please clarify.

Response:

Thank you for this observation, we apologise as this should have been included in the Appendix originally submitted. We have added the steam inhalation section to the Appendix on page 2, and it reads as follows:

“Have you previously used steam inhalation? How do you feel about it?

Prompt: offer an explanation if participant unsure.

What made you decide to use steam inhalation? Did you receive any advice?

How often did you use it? (e.g. regularly, sporadically).

If you stopped using it, why? (e.g. severity of symptoms settling, otherwise coping?)

If you haven’t used steam inhalation, why? Is this something you would try in the future, if so why (not)?”

Analysis

Comment:

Please state in the manuscript whether the analysis was conducted using any software.

Response:

Thank you for this suggestion, we have edited the start of the analysis section as follows on page 4. “Inductive thematic analysis[15] was conducted by hand on all transcripts to determine factors which influence patients’ decision to consult a GP or use an alternative treatment for RTI, as well as being open to themes outside of the core aims of the study.”

Comment:

“28 There is some repetition in the results.”

Thank you for making the changes outlined. In addition, line 13 “However, there was some confusion reported as to the role of delayed prescribing” is repeated on line 24. I suggest deleting line 13.

Response:

Thank you for this comment, this first paragraph under Theme 5 has been edited as suggested. It now reads as follows on page 9: “Delayed prescribing was generally viewed as a positive technique and patients reported being happy to accept whichever method of delayed prescribing the GP recommended. None of the participants expressed disappointment at not receiving an antibiotic prescription for immediate use. Delayed prescribing was generally viewed by participants as a positive technique which could give them reassurance of having access to a prescription just in case. The main benefit was seen as the patient having some decision in their own treatment while preventing them from unnecessarily taking antibiotics.”

Discussion

The following two comments are addressed together:

- Line 7-8, page 13. The finding that “beliefs relating to whether they thought they had caught the same condition as other people they knew” is not presented in the results. Please either add to the results or remove from here.
- “35 Implications for clinical practice, paragraph 3: how was accuracy of illness severity (“more often incorrect than correct”) assessed in this qualitative study? Please clarify if this sentence relates to results from the main trial. If the focus is on valuing participants’ own perceptions of severity, I am unclear how their perceptions of the severity of their symptoms can be deemed “more often incorrect than correct”. Please clarify how this judgement has been made?

Response:

Thank you for these comments, we agree that there needs to be greater clarity here. As such, we have edited these sentences and they now read as follows on page 12: “This qualitative study shows how patients’ decisions to consult a GP were often related to perceived indicators of illness severity.”

Comment:

The use of steam inhalation is not discussed in detail in this section. What conclusions do the authors draw in relation to implications for clinical practice for this treatment option in relation to the literature?

Response:

Thank you for this observation, we acknowledge that it should have been included. We have edited the second paragraph in the ‘Implications for clinical practice’ section on page 12, which now reads as follows: “Regarding steam inhalation, results from the PIPS trial[13] showed no impact on symptoms and minor scalds in some individuals; combined with the findings presented in this paper (mostly short term help at best) and prior literature[25] means that until more robust data are available, advice to use steam cannot be justified. More frequent/prolonged use of steam might help, but this would need to be shown before a recommendation could be made.”

VERSION 3 – REVIEW

REVIEWER	Joanna May Kesten University of Bristol, NIHR CLAHRC West and NIHR Health Protection Research Unit
REVIEW RETURNED	Thank you for responding to my comments. The clarity of the manuscript is much improved. I don't believe any further amendments are required prior to acceptance for publication.